# The association between physical complications following female genital cutting and the mental health of 12-year-old Gambian girls: A community-based cross-sectional study

Bothild Bendiksen[1]*, Trond Heir[1,2], Fabakary Minteh[3], Mai Mahgoub Ziyada[1,2], Rex A. Kuye[3], Inger-Lise Lien[1]

1 Norwegian Centre for Violence and Traumatic Stress Studies, Oslo, Norway, 2 The University of Oslo, Institute of Clinical Medicine, Oslo, Norway, 3 Department of Public & Environmental Health, School of Medicine & Allied Health Sciences, University of The Gambia, Brikama Campus, The Gambia

* b.a.bendiksen@nkvts.no

## Abstract

### Background

Female genital cutting (FGC) involve an acute physical trauma that hold a potential risk for immediate and long-term complications and mental health problems. The aim of this study was to examine the prediction of depressive symptoms and psychological distress by the immediate and current physical complications following FGC. Further, to examine whether the age at which 12-year-old Gambian girls had undergone the procedure affected mental health outcomes.

### Method

This cross-sectional study recruited 134 12-year-old girls from 23 public primary schools in The Gambia. We used a structured clinical interview to assess mental health and life satisfaction, including the Short Mood and Feeling Questionnaire (SMFQ), the Symptom check list (SCL-5) and Cantril's Ladder of Life Satisfaction. Each interview included questions about the cutting procedure, immediate- and current physical complications and the kind of help and care girls received following FGC.

### Results

Depressive symptoms were associated with immediate physical health complications in a multivariate regression model [RR = 1.08 (1.03, 1.12), p = .001], and with present urogenital problems [RR = 1.19 (1.09, 1.31), p < .001]. The girls that received medical help following immediate complications had a lower risk for depressive symptoms [RR = .73 (.55, .98), p = .04]. Psychological distress was only associated with immediate complications [RR = 1.04 (1.01, 1.07), p = .004]. No significant differences in mental health outcomes were found between girls who underwent FGC before the age of four in comparison to girls who underwent FGC after the age of four.

**Data Availability Statement:** All relevant data necessary for understanding the content presented in this manuscript are included in the text and tables. Data are from the project "Physical and

psychological healthcare for girls and women with Female Genital Mutilation/Cutting (FGM/C)". According to the approval from the Norwegian Regional Ethics Commmittee and the Gambia Government/ MRC Joint Ethics Committee, the data is to be stored properly and in line with the Norwegian and the Gambian Law of privacy protection. Public availability would compromise privacy of the respondents. However, anonymised data is freely available to interested researchers upon request, pending ethical approval from our Ethics committees: ethics@mrc.gm/ post@helseforskning.etikk.no. The project leader, Inger-Lise Lien (i.l.lien@nkvts.no) may also be contacted with a request for the data underlying our findings.

**Funding:** The project received funding from the Norwegian Research Council (NFR): Grant Number 262757. https://www.forskningsradet.no/en/ The funders had no role in the study design, data collection and analysis, decision to publish, or preparation of the manuscript.

**Competing interests:** The authors have declared that no competing interests exists.

## Conclusion

Our findings indicate that the immediate and long-term complications following FGC have implications for psychological health. Only a minimal number of girls received medical care when needed, and the dissemination of health education seems crucial in order to prevent adverse long-term physical and psychological health consequences.

## Introduction

Female genital cutting (FGC) consists of all procedures involving partial or total removal of external female genitalia or other injury to the female genitalia for cultural, religious or other nontherapeutic reasons. The World Health Organization classification [1] differentiates between four types of FGC: Type I, Type II, Type III and Type IV. In the Republic of The Gambia (The Gambia), the latest estimates report a prevalence rate of 76.3% [2, 3]; Type I (clitoridectomy) accounts for 66% of all FGC cases, Type II (excision) accounts for 26% and Type III accounts for less than 10% [3, 4].

Over the past three decades, there has been increasing evidence that FGC harms the physical health and well-being of girls and women. The immediate medical complications of FGC include excessive bleeding, pain, genital tissue swelling, urine retention and problems with wound healing [5, 6]. Long-term or chronic complications, including genitourinary problems, infections and a wide range of sexual and obstetric complications, have consistently been reported [2, 4, 7–16].

Although the adverse psychological and psychosocial impacts of FGC have been less studied, depression, somatization, post-traumatic stress disorder (PTSD) and symptoms of anxiety, distress, insomnia, helplessness, anger, irritability and low self-esteem are among the mental health problems reported by studies among female immigrants [17], women residing in their countries of origin [18, 19] and among children and adolescents [20, 21].

Moreover, FGC involve an acute physical trauma that is likely to be associated with both immediate and long-term genital pain. Physical injuries hold a potential risk for post-injury mental health problems [22], where pain following a traumatic injury has been linked to depressive symptoms in children and adolescents [23].

Typically, FGC is performed during a specific time-span that varies from one country to another. In The Gambia, traditional female circumcisers usually perform the procedure between 7 days after birth up to preadolescence [2]. To date, only one study has addressed the significance of the age at which a girl undergoes FGC, which is relevant for the level of recollection, which, in turn, might affect an individual's psychological health and well-being. Among immigrant women in the Netherlands who had undergone FGC, Knipscheer et al. found that a vivid recollection of the FGC event had an adverse impact on mental health and well-being [24].

The aim of this study was to examine the prediction of depressive symptoms and psychological distress by the immediate and current physical complications following FGC. Further, to examine whether the age at which 12-year-old Gambian girls had undergone the procedure affected mental health outcomes.

## Methods

### Participants

This paper presents data from a cross-sectional community-based sample of 134 12-year-old girls who had undergone FGC. The participants in this study were drawn from a larger sample

of 300 girls from 23 public primary schools in the urban areas of The Gambia. Of the sample of 300 girls, 251 (84%) accepted the invitation to participate in the study and 49 (16%) declined. Thus, the inclusion criteria in this present study sample were 12-year old schoolgirls form urban areas of the Gambia who had undergone FGC, who were willing to participate and of whom parents gave their written consent (n = 134). Girls who had not undergone FGC (n = 115), had poor English language skills, or had missing data on mental health measures (n = 2), the Short Mood and Feeling Questionnaire (SMFQ), the Symptom check list (SCL-5) and Cantril's Ladder of Life Satisfaction (Expected Life Satisfaction, ELS), were excluded from the analyses.

## Procedure

The study sites were lower basic schools in the Kanifing region and chosen in collaboration with the Regional Education Directorate, Region One, which have an high school enrolment rate and ethnic mixed communities. The Gambia has a state educational system inherited from the British, where English is still the official language for government and education. The data were collected over a three-week period from the end of January to early February 2018. Pilot-testing of the questionnaire was conducted on 12-year-old girls (n = 6) from a lower basic school in this same area. A collaborative research team, comprised of professors, lecturers and faculty members and graduate students at the University of The Gambia, School of Medicine and Allied Health Sciences/ Department of Public and Environmental Health, was involved in the planning, training and implementation of this study.

The headmasters and their assistants (usually female teachers) at each school received detailed information about the study before they relayed the study information to all the 12-year-old female students and to the girls' parents, with particular emphasis on the confidentiality, anonymity and voluntary nature of participation. A female teacher provided the information to the 12-year old girls who were gathered in groups. Parents received oral information from the headmaster either by attending an information meeting held at the school or by phone. Additionally, all parents received written information through the consent form. Those parents who were illiterate (about 50%) received oral information before signing the consent form with a fingerprint.

An experienced researcher and child psychiatrist was responsible for providing a three-day training session for the research assistants and supervisors on how to conduct a structured interview, including the questionnaire and checklists. In addition, a lecture was held by a professor at the Department of Public and Environmental Health, who identified and translated potentially difficult technical terms and concepts in English language into the different local languages.

Trained female students at the University of The Gambia collected the data. The interviewers were trained to build a confident questionnaire-building report between the interviewer and the interviewee regarding the secrecy and the privacy so that girls could feel safe regarding the information they gave. The interviewers made clear for girls that there would be no social stigma or legal consequences. Assessment of the 12-year-old girls was based on a 40-minute clinical interview, which took place in an office or a classroom at their school, undisturbed and away from other students and teachers. All interviews were conducted in English, and enquiries and uncertainties were explained in the local languages.

## Measures

The structured interview began with general sociodemographic questions, such as age, number of siblings and ethnic group affiliation. Further, the following questions pertaining to FGC

were asked: -"Are you circumcised/ cut?" (yes = 1, no = 2, I am not sure if I am cut = 3) (*selection variable*).

For those girls who confirmed that they were cut, additional questions about FGC and its circumstances were asked (*exposure/grouping variables*): i) If you are circumcised/ cut, at what age were you circumcised/ cut? (before 4 years of age = 1, between 4 years and 7 years = 2, after 7 years = 3). ii) Who did the circumcision/ cutting? A circumciser, a medical doctor, a nurse/ midwife, other person (each with answer categories yes = 1, no = 0). Iii) Describe the effect of the circumcision/ cutting on your body immediately after it happened with the following answer categories: it was very painful, I was bleeding, it was difficult to urinate, I could not urinate, I got a fever, I had problems walking and sitting down (each question with answer categories yes = 1, no = 0). iv) Describe the effect of the circumcision/ cutting on your body today: do you have pain in your private parts/ around the circumcision/ cutting today, do you spend long time urinating, do you have pain when you urinate (each with answer categories yes = 1, no = 0). v) Further, we asked if girls needed any extra help after the circumcision/ cutting. If the answer was "yes", we asked the following questions: were you taken to a health facility, were you taken to a marabou, did your mother take you to an imam, was the circumciser called, were you taken elsewhere with your problem (score no = 0, yes = 1, for each question). Open labelled questions about what kind of help they received were asked, where traditional help mostly were herbs, ointment, juju or prayers. We also asked if the circumcision/ cutting healed without any problem (yes, no) in order to validate the information of their reports on immediate complications and need for help.

We did not ask specific questions about the type of FGC and did not verify FGC by physical examination. The questionnaires on mental health and well-being are described below (*outcome measures*).

The SMFQ is a shortened version of the 33-item Mood and Feeling Questionnaire (MFQ) [25]; it was developed as a way to obtain a brief evaluation of depressive symptomatology in children and adolescents 8–18 years of age [26, 27]. The psychometric properties have been found to be reliable across both clinical and community samples [28–32] and in different ethnic groups [33, 34]. The SMFQ consists of 13 items with total scores ranging from 0–26. Each of the 13 items is rated on a 3-point scale ranging from 0–2 (0 = not true, 1 = sometimes true, 2 = true). Thapar et al. demonstrated a cut-off point of 11 to show high sensitivity and specificity in a sample of children aged 8–16 years [29]. In the sample in our study, we found a left-distribution of the SMFQ scores where participants with a score of 11 represented the top 6% of the sample. The internal consistency of the scale was found to be satisfactory (Cronbach's $\alpha$ = 0.71).

We used the SCL-5, a widely used short version of the Hopkins Symptom Check List-25 (HSCL-25) [35], to assess symptoms of psychological distress. The SCL-5 scores strongly correlate with the original HSCL-25 scores, where r = .91 [36], and it has shown good reliability, good specificity (82%) and sensitivity (96%) [36, 37]. The SCL-5 has been validated in samples of youths [36, 38]. In the present sample, the internal consistency was lower than previously demonstrated (Cronbach's $\alpha$ = 0.66).

The SCL-5 includes three questions about anxiety ("feeling fearful/ frightened/ scared", "nervousness or shaking inside", "worrying too much about things") and two questions about depression ("feeling hopeless about the future", "feeling depressed/ blue"). Each question is rated on a 4-point scale ranging from 1–4 ("not at all", "a little bit", "quite much", and "extremely"). Thus, the SCL-5 has a maximum total score of 20. The cut-off point is usually set at 2 or more for the individual scores and at 10 for the total score, which, in the sample in the present study, represented the top 20% indicating clinical levels of distress. We found a left-skewed distribution of the SCL-5, where 20% of the sample had a score above 10 (at the 75th percentile).

We used a single-item questionnaire, Cantril's Ladder of Life Satisfaction [39], to measure both current life satisfaction (CLS) and expected life satisfaction (ELS) (imagined life-

satisfaction in 2 years). A vertical ladder was drawn and shown to the students who were informed that the scores ranged from "the worst possible life imagined" (1) to "the best possible life imagined" (10). Good reliability and convergent validity with other emotional measures have been demonstrated across samples of 11–15 year-old children [40]. A binary cut-off for "high life satisfaction" ($\geq 6$) and "low life satisfaction ($<6$) was used by Currie et al. in their Health Behaviour in School-aged Children (HBSC) study [41].

## Statistical analyses

The sociodemographic characteristics include the means and standard deviations (SD) for age and the number of siblings for the young female students who had undergone FGC. The affiliation to ethnic group, the age at which girls underwent FGC, the immediate and long-term physical health complications and type of health care are given in numbers and percentages. Moreover, medians and range are presented for age, number of siblings and the immediate and current physical health complications.

We conducted Poisson regression analyses, unadjusted and adjusted, to examine the relationship between the number of immediate and present physical complications and mental health outcomes. We adjusted for age, number of siblings, ethnic group affiliation (Mandinka/other) and the age in which the FGC procedure was conducted (before or after the age of 4). We also adjusted for the type of help the girls received following complications due to FGC. We categorised the type of help provided as follows: i) traditional (called for the circumciser), ii) religious (contact with a marabou or an imam) and iii) medical help (taken to a health facility). In the multivariate analyses, we examined the data for possible interactions between immediate and current physical complications, as well as between the physical complications and other covariates in the model. Crude and adjusted Incidence Risk Ratio (IRR) with 95% confidence intervals (CI) were given using robust sandwich estimator of variance.

The main outcome variables in this sample were skewed and clustered to the left: SMFQ: Mean = 5.21 (4.01) and SCL: Mean = 8.26 (2.56). We used Mann-Whitney U Test to compare the level of depressive symptoms (SMFQ), the level of psychological distress (SCL-5) and the CLS and ELS for the participants who had undergone FGC before-and after the age of 4.

The significance level was set at p = 0.05 (two-sided). The two-sided p-values were not corrected for multiple testing. The data were processed using IBM SPSS statistics version 25.

## Ethics

This study was approved by the Norwegian Regional Ethics Committee, number 2017/997 and The Gambia Government/MRC Joint Ethics Committee, number R017 034, and it was conducted accordingly. Consistent with the ethical clearance, return of the fingerprinted consent forms from the illiterate parents after providing them with comprehensive oral information about the project was considered to be informed consent. Furthermore, a contingency plan was set up to deal with potential serious reactions among the study participants. The plan consisted of five points to safeguard the girls that were interviewed. This included using professional and trained interviewers, conducting the interviews in safe settings, providing contact information about people with mental health competence and securing a signed collaboration agreement with Faji Kunda Health Centre.

## Results

All the participants (n = 134) were Muslims and mainly affiliated with the Mandinka ethnic group (n = 84, 62.7%). Other ethnic groups included Fula (n = 22), Wolof (n = 11), Jola (n = 9), Serer (n = 3), Bambara (n = 2), Sarehule (n = 2) and Jahanka (n = 1).

Table 1 shows the background characteristics of the participants, the prevalence of immediate and present physical health complications and help received following immediate complications. The age at which girls underwent FGC did not differ between the Mandinka and the other ethnic groups (i.e. before or after the age of 4).

## Immediate and current health complications associated with mental health

Nearly all the girls who remembered the event and its circumstances reported of immediate physical complications following FGC (Table 1). The complications included bleeding (n = 78, 91%), difficulties urinating (n = 79, 92%), severe pain (n = 85, 99%), fever (n = 46, 53%) and problems walking (n = 68, 79%) and sitting down (n = 58, 67%). Sixty girls (45%) reported five or more immediate complications.

The girls who reported current physical complications following FGC (n = 35, 26.1%) said that they spent a long time urinating (n = 26, 19%), suffered from pain when urinating (n = 22, 16%), or they had pain in their genital area (n = 11, 8%).

The immediate and present physical health complications were both significantly associated with depressive symptoms, while only the number of immediate complications was related to psychological distress after the adjustments (Table 2). In the adjusted model, one additional unit increase in the number of immediate physical health complications increased the mean number of depressive symptoms by 6% and psychological distress by 4%. The number of current/long-term physical complications increased the mean score of depressive symptoms by 19%, whereas the effect on psychological distress was non-significant.

## Traditional help and healthcare for complications from FGC

Of the girls who remembered the procedure, 60% reported that they needed help after experiencing immediate complications from the procedure (Table 1). Receiving traditional help from the circumciser or a religious leader following immediate complications was not associated with any mental health outcomes, while the girls taken to a healthcare facility had a lower risk for depression (Table 2).

## Age at which girls were subjected to FGC

There were no significant differences in levels of depressive symptoms or psychological distress among the girls that were subjected to FGC before or after the age of 4. The CLS and ELS scores were also similar in the two groups (Table 3).

**Table 1. Sample characteristics of the 134 girls subjected to FGC.**

|  | M (SD) | Median | Range |
|---|---|---|---|
| Age (months) | 149.26 (6.54) | 148.50 | 66 |
| Number of siblings | 4.43 (2.57) | 4 | 15 |
| Number of immediate physical complications | 3.43 (3.09) | 4 | 9 |
| Number of present physical complications | 0.46 (0.87) | 0 | 3 |
|  | Yes, N (%) | No, N (%) | No remembrance, N (%) |
| Age of the FGC procedure ≥ 4years | 60 (44.8) | 74 (55.2) | - |
| Ethnic group affiliation (Mandinka vs *other) | 84 (62.7) | 50 (37.3) | - |
| Immediate physical complications following FGC | 82(61.2) | - | 52 (38.8) |
| Help received (circumciser) | 35 (26.1) | 53 (39.6) | 46 (34.3) |
| Help received (religious leader) | 22 (16.4) | 66 (49.3) | 46 (34.3) |
| Help received (health facility) | 11 (8.2) | 77 (57.5) | 46 (34.3) |
| Current physical complications following FGC | 35 (26.1) | 95 (70.9) | 4 (3) |

**Table 2. The associations between mental health outcomes and the number of immediate- and current physical health complications following FGC.**

| | | Depressive symptoms (SMFQ score) | | | | | |
|---|---|---|---|---|---|---|---|
| N = 134 | *scale* | *Crude IRR* | *95%CI* | *p* | *Adj. IRR* | *95%CI* | *p* |
| Number of siblings | 0–15 | .93 | .90, .97 | .000 | .93 | .90, .97 | .000 |
| Ethnic group: Mandinka | yes | .83 | .71, .96 | .01 | .88 | .74, 1.06 | .17 |
| Immediate physical complications | 0–9 | 1.07 | 1.03, 1.11 | .001 | 1.06 | 1.02, 1.11 | .002 |
| Help received (health facility) | yes | 0.75 | 0.56, 1.00 | .05 | 0.73 | 0.55, 0.98 | .04 |
| Current physical complications | 0–3 | 1.21 | 1.11, 1.32 | .000 | 1.19 | 1.09, 1.30 | .000 |
| | | Psychological distress (SCL-5 score) | | | | | |
| N = 134 | *scale* | *Crude IRR* | *95%CI* | *p* | *Adj. IRR* | *95%CI* | *p* |
| Number of siblings | 0–15 | .97 | .94, .99 | .02 | .97 | .94, .99 | .02 |
| Immediate physical complications | 0–9 | 1.04 | 1.01, 1.08 | .01 | 1.04 | 1.00, 1.07 | .03 |
| Current physical complications | 0–3 | 1.08 | 1.00, 1.17 | .05 | 1.08 | 0.99, 1.61 | .06 |

Note: IRR = Incidence Risk Ratio. Adjusted for age, number of siblings, ethnic group affiliation (Mandinka/ other), age of FGC procedure (≥4>) and help received after immediate complications following FGC: i) called the circumciser, ii) talked with a religious leader, iii) taken to a health facility.

## Discussion

In this study, we examined whether the number of immediate and current physical complications following FGC were associated with depressive symptoms or psychological distress in 12-year-old girls subjected to FGC. Furthermore, we examined whether the age at which the girls underwent FGC was associated with mental health outcomes and well-being.

For the girls who were able to recall the circumstances, nearly all reported immediate physical problems following FGC. One-quarter of the girls reported current urogenital health problems. Immediate physical complications following FGC were related to higher levels of depressive symptoms and psychological distress, while current problems were significantly related to more depressive symptoms. More siblings seemed to a protective factor, while the affiliation to ethnic group was not associated with the mental health outcomes. Traditional help following immediate complications was not related to mental health outcomes, while a lower risk for depressive symptoms was found for the girls who were taken to a healthcare facility when complications became evident. The levels of depressive symptoms, psychological distress and CLS and ELS were similar in the girls who underwent FGC before age 4 in comparison to girls who were subjected after 4 years of age.

In line with reports from the systematic review conducted by Berg et al. [5], in our study the immediate physical complications that were reported were bleeding, pain and difficulties urinating. More than 50% of the girls had a fever, which may indicate that they got an infection following the procedure. More than 90% of the girls who had a recollection of the event reported having one or more complications. Our findings exceed the results in previous reports of the proportion of females suffering from one or more complications [6, 42, 43], which might reflect the way the questions were asked (i.e. we asked if the complication

**Table 3. Symptom scores for depression, psychological distress and life satisfaction for participants who underwent FGC before or after the age of 4.**

| | FGC procedure before the age of 4 | FGC procedure after the age of 4 | | | | |
|---|---|---|---|---|---|---|
| | *Median (n)* | *Mean (n)* | | *U* | *z* | *P* |
| **Depressive symptoms** | 4 (74) | 5 (60) | | 1893 | -1.47 | .14 |
| **Psychological distress** | 7.5 (74) | 8 (60) | | 1956 | -1.191 | .23 |
| **Current Life satisfaction** | 5 (74) | 6 (60) | | 1959 | -1.192 | .23 |
| **Expected life satisfaction** | 8 (74) | 9 (60) | | 1915 | -1.394 | .16 |

occurred, but we did not ask about the severity of the complications). Our findings clearly contrast the results from a recent study that used a convenience sample from a gynaecological clinic in Saudi Arabia [44], which indicated that less than 3% of the women who had undergone FGC reported two or more immediate complications following the procedure. The proportion of girls who suffer and the type of immediate complications seem to vary across studies depending of the degree to which the respondents remember the circumstances under which the procedure was performed, as well as the scope of the procedure [4, 6, 9, 42, 43]. FGC Type I accounts for two-thirds of all the FGC cases in The Gambia, while infibulation is performed in less than 10% of the cases [2, 4]. Although infibulation might have a greater risk of complications [6], our data support that clitoridectomy and excision also represent a substantial risk for adverse immediate physical health consequences.

About 25% of the girls in our study reported suffering from current urogenital complications, mainly genital pain and urinary problems, which is in line with a clinical study of 871 girls and women from The Gambia conducted by Kaplan et al. [4]. Long-term urogenital complications are well documented for all types of FGC [4, 6, 9, 11, 16, 19, 42, 45]. Although not confirmed by the systematic review conducted by Berg et al. [6], studies have reported abnormal scarring, inclusion cysts and labial fusion in girls and women following FGC Type I and Type II [4, 15, 42, 45]. In pre-pubertal girls, low oestrogen levels may cause labial fusion, inflammation and recurrent urinary tract infections (UTI) [46]. Thus, the anatomical changes following FGC put younger girls at a higher risk for ascending urogenital infections [42].

The bulk of evidence on complications following FGC pertains to sexual and obstetric consequences in adults [47–49], while children and adolescent girls have received less attention. Most of the studies focusing on children are from clinics, and they describe very serious complications [13, 15, 42, 45, 50]. However, urogenital complaints of varying severity might be impairing for the affected girls and further impact their reproductive physical health and their mental health and well-being.

Only a few studies have provided documentation on the psychological impact of FGC on children and adolescents. A cross-sectional clinical study from Egypt found significantly more psychological problems, including anxiety and depression, among 14–19-year-old females who had undergone FGC in comparison to those who had not [20]. A community-based study from Northern Iraqi found that 10–16-year-old Kurdish girls subjected to FGC had higher rates of PTSD, depression, anxiety and somatoform disturbances than girls who had not undergone the procedure [21]. The participants represented in the Egyptian sample were recruited from an outpatient gynaecological clinic; thus, they constitute a group of help-seeking young females [20], but the authors did not address whether the psychological outcomes were related to the level of gynaecological problems. Likewise, the Iraqi study did not provide information on somatic complaints due to FGC [21]. In a systematic review conducted by Berg et al. [49], the authors argued that the evidence on the impact of FGC on psychological health is too sparse and of low quality to draw any firm conclusions. Although multiple factors may influence mental health outcomes among females who have undergone FGC, it seems reasonable to assume that the extent of physical injury due to the procedure and its consequences will eventually have an impact on mental health and well-being for this group.

Interestingly, the number of siblings was found to have a protective effect against depression and psychological distress. We can only speculate, but girls who have older sisters, unlike an only child, might have an opportunity for shared experiences of the circumcision ritual, thus reduce the risk for mental health consequences.

FGC is a mutilating physical trauma, which, in addition to the immediate pain, might also be associated with long-term genital pain. Children exposed to various physical injuries may be at risk of post-injury mental health problems [22, 51–56]. Different characteristics of the injury might contribute to an individual's emotional responses. The perception of fear,

helplessness and horror at the time of the injury, as well as dealing with the pain and other bodily consequences, may elicit individual differences in psychological reactions. FGC is a traumatic disfiguring injury, and the World Health Organization has characterised it as mutilation [57]. Mutilating injuries following a physical trauma have been linked to acute stress and post-traumatic stress as well as anxiety and depressive symptoms in children and adolescents [54]. While the pain associated with a physical traumatic injury is mostly acute and transient, it is long remembered. In both physical injury and chronic illness, pain has been found to be associated with a substantial likelihood of developing mental health problems, including depressive symptoms, anxiety and psychological distress [58–61].

In our study, the vast majority of girls reported that they needed help because of immediate medical complications. A lower risk for depression was found in girls who received medical help, while no associations were found between traditional help and mental health outcomes. However, less than 10% of the girls were taken to a healthcare facility. In most cases, the circumciser was contacted, while some of the girls were taken to a religious leader. Our finding of extremely low health service utilisation is in line with the findings reported in a study on FGC from Sierra Leone conducted by Bjälkander et al. [7]. Further, a clinical study of injuries among children and adolescents in Malawi reported that health service utilisation was less than 10% [62]. Medical treatment may be too expensive and access to healthcare services may be limited. Secrecy pertaining to FGC could be another factor contributing to not seeking healthcare. In The Gambia, a statutory prohibition against FGC was passed in 2015 [63]. Even though most of the girls in our sample underwent the procedure before 2015, The Gambia has a long history of awareness campaigns against this practice [64–69].

Mental health problems following FGC may vary based on the age at which a girl underwent the procedure, where the level of remembrance might be crucial for the interpretation of the event. In humans, the earliest autobiographical memories date from between the ages of 3 and 4 [70, 71]. Therefore, it is not surprising that, in our sample, 16% of those who underwent the procedure before the age of 4 had a recollection of the procedure and its circumstances. The literature on the potential adverse psychological consequences of FGC mainly lack reports on the impact of at which age the girls were subjected to the procedure and the level of remembrance. Although FGC was performed before the age of 4 in more than half of the girls in our Gambian sample, we found no differences in mental health outcomes or life satisfaction for the girls who underwent FGC before the age of 4 and those who underwent it after this age. In contrast, Knipscheer et al. found that a vivid recollection of the circumcision was a significant risk factor related to mental health problems among immigrants who had undergone FGC residing in the Netherlands [17, 24]. Similar to The Gambia, the procedure is performed before the age of 5 in about half of the countries practicing FGC [72]. Although the cutting occurs at a later age in countries like Iraqi Kurdistan, Somaliland and Egypt [21, 72–74], reports suggest a general downward shift in the average age at the time of cutting [75, 76]. The reasons for this downward shift might be that younger girls are less resistant than older girls are, and that FGC can be done more discretely in younger girls. In their study from Somaliland, Powell et al. suggested that fundamental shifts in the practice towards a lower cutting-age of girls, increased medicalisation and less invasive cutting might represent a need to find an acceptable balance between cultural expectations and health preservation [75]. To date, no changes in the prevalence or trends of FGC has been observed in The Gambia [72].

## Strengths and limitations

This study has some limitations. First, we were unable to obtain information about socioeconomic background at the individual level. Mental health problems have been found to be associated with lower socioeconomic status (SES) and lower educational levels [77, 78], which

might have been possible confounders in the current study. However, we argue that the lack of sociodemographic data did not affect the credibility of the results as all the participants were enrolled in public schools in the urban areas, which resulted in a balanced sample as specified by low-to-moderate level of income and education. Moreover, among young females who have undergone FGC, sociodemographic characteristics have been shown to be less important for mental health outcomes than the exposure itself [20].

A second limitation of the study is that the clinical interviews consisted of instruments developed in Western countries, and they may not be psychometrically, linguistically or culturally adapted for use in other settings. However, the extensive three-day training of the female university students who collected the data included a comprehensive explanation on the meaning of the different terminologies and concepts, and their equivalence in the various local languages. The instruments included in the structured interview were previously validated among children or youth [27–29, 38] and among different ethnic groups [33, 34]. The internal consistency for SCL-5 was not entirely acceptable, while SMFQ showed moderate reliability.

Finally, information was based on self-report of FGC and its complications. Studies have reported some inconsistencies in agreement between self-reported and observed FGC status [79, 80]. None of the included participant reported being uncertain about their FGC status although they were given the opportunity for this answer. We thoroughly reviewed the data for inconsistencies of the answers on questions pertaining to the FGC procedure, complications and help received. Sixteen percent of the girls who underwent FGC before age four gave some information of the event and its circumstances, which may represent a source for recall bias. On the other hand, the young age of participants in this sample reduced the influence of recall bias of the immediate health complications and the kind of help they received as they had the procedure and its circumstances very fresh in their memories.

Additional strengths of the study include the high participation rate and the community-based design.

## Conclusion

Despite their young age, the level of both present urogenital complaints and the immediate complications they faced following the FGC procedure posed a significant risk for depressive symptoms among girls in this study. Moreover, the immediate physical complications was associated with psychological distress, mainly. Only a minimal number of girls received medical care when needed and the dissemination of health education seems crucial in order to prevent adverse long-term physical and psychological health consequences.

## Acknowledgments

The authors thank the young female participants and their parents for consenting to their daughters' participation in a study on such a sensitive topic. We are very grateful to the headmasters and teachers at the participating schools who made it possible to conduct this study. We especially acknowledge the work of our wonderful collaborating research team at the University of The Gambia, the Department of Public & Environmental Health and the School of Medicine & Allied Health Sciences, whose female students performed the data collection, and their supervisors, who followed-up during training and the data collection process.

## Author Contributions

**Conceptualization:** Bothild Bendiksen, Fabakary Minteh, Mai Mahgoub Ziyada, Rex A. Kuye, Inger-Lise Lien.

**Data curation:** Bothild Bendiksen, Inger-Lise Lien.

**Formal analysis:** Bothild Bendiksen.

**Funding acquisition:** Mai Mahgoub Ziyada, Inger-Lise Lien.

**Investigation:** Bothild Bendiksen, Fabakary Minteh.

**Methodology:** Bothild Bendiksen, Trond Heir, Fabakary Minteh, Inger-Lise Lien.

**Project administration:** Rex A. Kuye, Inger-Lise Lien.

**Resources:** Rex A. Kuye, Inger-Lise Lien.

**Supervision:** Trond Heir.

**Validation:** Bothild Bendiksen.

**Writing – original draft:** Bothild Bendiksen.

**Writing – review & editing:** Bothild Bendiksen, Trond Heir, Fabakary Minteh, Mai Mahgoub Ziyada, Rex A. Kuye, Inger-Lise Lien.

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
