## [Decision Letter · Decision Letter 0]

22 Sep 2020

PONE-D-20-03360

Physical complications following female genital cutting affect the mental health of 12-year-old Gambian girls: A community-based cross-sectional study

PLOS ONE

Dear Dr. Bendiksen,

Thank you for submitting your manuscript to PLOS ONE. After careful consideration, we feel that it has merit but does not fully meet PLOS ONE’s publication criteria as it currently stands. Therefore, we invite you to submit a revised version of the manuscript that addresses the points raised during the review process.

We look forward to receiving your revised manuscript.

Kind regards,

Kimiyo Kikuchi

Academic Editor

PLOS ONE

Journal Requirements:

2.We note that you have indicated that data from this study are available upon request. PLOS only allows data to be available upon request if there are legal or ethical restrictions on sharing data publicly. For information on unacceptable data access restrictions, please see http://journals.plos.org/plosone/s/data-availability#loc-unacceptable-data-access-restrictions.

Additional Editor Comments (if provided):

1. Introduction

L73-77: The objective of the introduction is redundant and is not consistent with that of the abstract.

2. Methods: what are the criteria of the participants?

3. L90-92: The participants’ attributes date should be described in the results.

4. L115-116: What are the questions pertaining to FGC? Please add the references if there are.

5. L133-134: Please write the references of HSCL-25 as well as that of SCL-5.

6. L143-144: Please add the references of “the cut-off point 2 or more for the individual scores and at 10 for the total score”.

7. L152: “samples of 11—5 years-old children”. Is this correct?

8. L170: Relative risk is not usually used in the cross-sectional study. Isn’t it an odds ratio?

9. L174: Why the age of 4 was a cut-off point?

10. Please add the Ethical committee approval number both for the Norwegian regional ethics committee and the Gambian government/MRC joint ethics committee.

11. L118: what are the contents of physical complications?

12. Table 2: Perhaps some of the current physical complications are related to past complications (immediate physical complications)? If so, the author has to remove one of the variables from “immediate physical complications” and “current physical complications”, as there should be an overlap.

13. Table 2: Why did the author not adjust with the variables “help received” in the analysis?

14. L164-168: How did the author select those variables? Please describe the criteria of the selection to justify that the author did not pick them intentionally.

15. L249-251: “more than 50% had a fever”, “more than 90% of the girls had complications”. Are they Gambian prevalences or worldwide prevalences?

Reviewers' comments:

Reviewer's Responses to Questions

**Comments to the Author**

1. Is the manuscript technically sound, and do the data support the conclusions?

Reviewer #1: Yes

Reviewer #2: Yes

2. Has the statistical analysis been performed appropriately and rigorously? 

Reviewer #1: Yes

Reviewer #2: No

3. Have the authors made all data underlying the findings in their manuscript fully available?

Reviewer #1: No

Reviewer #2: Yes

4. Is the manuscript presented in an intelligible fashion and written in standard English?

Reviewer #1: Yes

Reviewer #2: Yes

5. Review Comments to the Author

Reviewer #1: The authors present findings for the research study titled “Physical complications following female genital cutting affect the mental health of 12-year-old Gambian girls: A community-based cross-sectional study”. Although the findings add towards the evidence of the impact of FGC on mental health on girls in Gambia, I find gaps in methods for it to be published as is. The following are the major and minor comments towards the manuscript.

Title: Physical complications following female genital cutting affect the mental health 1 of 12-year-old Gambian girls: A community-based cross-sectional study. I suggest modification to the title because cross-sectional studies do not depict causality.

The Abstract is well written and reads well.

Introduction

i. Include a description of who performs the FGC on girls in Gambia

Methods Section

i. It will be helpful to describe in details the study sites/settings especially the schools

ii. Describe in details the characteristics of education system in Gambia, especially because interviews were conducted in English for girls 12 years old?

iii. How did the authors obtain information regarding FGC status and the complications they suffered then, as well as current complications?

iv. How would a girl 12 years who was cut may be at 1 month or when younger than 4 years old remember the incidence?

v. Did the authors had methods of validating the information presented by the girls

vi. How did you address recall bias that is widely documented in regard to FGC

vii. How was sampling done?

viii. It would be helpful to describe what the traditional healers and religious persons did in supporting of the girl following the FGC-related complications. Again how would the young girl recall?

Results

i. Page 10 line 215-218: Receiving traditional help from the circumciser or a religious leader following immediate complications was not associated with any mental health outcomes, while the girls taken to a healthcare facility had a lower risk for depression. This is an important point but under described in the discussion. Could it be the girls would think by being taken back to the healers they would again be done something more harmful? In the absence of information on who performs the cut I can only speculate.

Discussion

i. It will be helpful to the shorten it

Limitations

i. Shorten the limitation to a few sentences

ii. The conclusion draws from the result but can benefit from a re-look

Reviewer #2: Introduction

• It is better to focus on the topic

Methods

• Please mention the inclusion and exclusion criteria

• It is not clear how the final sample size is determined

• It is not clear how the schools were selected, Further description of the steps involved in the selection of the schools is needed

• Why do the authors select 12 years old girls as a study subject

• Pretest might be important for such questionnaire , have you done a pretest

• In the measures subsection section ,the authors pointed out that “we asked the girls whether or not they had been undergone FGC”

There is no need to ask this if the samples have undergone FGM. What is the intention of asking this question?

• There is no need to do mean , median, SD, IQR for the same variable, It depends on the presence of normal distribution, please check the normal distribution and do the required analysis accordingly

• Please discuss about whether the data fulfilled the assumptions of t-test before conducting it

Results

• It is not presented correctly, The concept mentioned in table 1 is written all over the result section

• The authors try to mention two separate concepts in table on

• The minimum and maximum values can be mentioned as single variable (range)

• 52 of the subjects didn't remember the complication, then how did 6 of them remember whether they received help or not

• it might be wrong to measure the presence of complication based on subjective data, Did the authors believe that the subjective data accurately represent the study objective

• Line 197,the authors mentioned that nearly all the girls who remembered the event and its circumstances (n = 86, 64%) while the value mentioned in table 1 is 82(61.2)

• the association with number of siblings is not clear, it is better to elaborate it in the discussion section

• what is the need of assessing the association of mandinka ethnic group with the outcome variables, what will be the recommendation if it showed the association

• Why do the authors consider age of 4 in comparing the mean value difference in the levels of depressive symptoms or psychological distress

Discussion

• It seems a literature review

• Please delete or elaborate concepts that are not presented in the result section. The discussion section should be written in line with the results

• Please try to compare the findings from various studies with your study

References

• Many references are old

6. PLOS authors have the option to publish the peer review history of their article (what does this mean?). If published, this will include your full peer review and any attached files.

Reviewer #1: **Yes: **Samuel Kimani

Reviewer #2: No

---

## [Author Response · Author response to Decision Letter 0]

6 Dec 2020

Response to reviewers Oslo, 06.11.2020

We would like to thank the academic editor and the reviewers for their valuable time and feedback. 

In this letter, we provide point-by-point response to both the editor and reviewers.

Response to the academic editor

Data sharing policy:

Response: Unfortunately, we have no approval for data sharing from the Regional Medical Ethics Committee, South-East Norway, or from The Gambia Government/MRC Joint Ethics Committee. We fully agree in the data sharing policy of PLOS ONE and will for future project from this department, apply for sharing the data.

1. Introduction

L73-77: The objective of the introduction is redundant and is not consistent with that of the abstract.

Response: The objective of introduction has been revised, page4, L75-78

2. Methods: what are the criteria of the participants?

Response: Revised in the method section, page 4, L84-88.

3. L90-92: The participants’ attributes date should be described in the results.

Response: Done. Page10, L267-269.

4. L115-116: What are the questions pertaining to FGC? Please add the references if there are.

Response: Questions pertaining to FGC are now included in the method section, page 6 and 7, L155-179.

5. L133-134: Please write the references of HSCL-25 as well as that of SCL-5.

Response: The references of HCL-25 and SCL-5 are already there, L193-194.

6. L143-144: Please add the references of “the cut-off point 2 or more for the individual scores and at 10 for the total score”.

Response: The references for SCL-5 cut-points are the same as for the SCL-5, Strand BH and colleagues, 2003, reference number 36 

7. L152: “samples of 11—5 years-old children”. Is this correct?

Response: I am sorry, this is a typo. Corrected in text, L223

8. L170: Relative risk is not usually used in the cross-sectional study. Isn’t it an odds ratio?

Response: I can see your point. Relative Risk (RR) is the same as Incidence Risk Ratio (IRR) used in Poisson and Negative bionomic regression analyses. These analyses may be used in a cross-sectional study if assumptions are not violated, i.e. the outcome variables are count data and not normally distributed. Logistic regression models with OR require a binary/ categorical outcome, which would been a possibility by using cut-points for SMFQ and SCL-5.

We have in the revised version used incidence risk ratio (IRR), which might be more appropriate, statistic section, L244-245, as well as in the result section, table 2, note, L327. 

9. L174: Why the age of 4 was a cut-off point?

Response: Age 4 was used as a cut-point as we tried to sort out the girls who had a recollection of the procedure. The questionnaire included three categories for when FGC happened: before the age of four, between 4 and 7 years, and after 7 years of age. Very few girls (n = 18, 13.4%) in this sample were cut after the age of 7. Still, about 25 of those girls who underwent the procedure before age 4 did remember being cut. Their answers regarding remembrance of the event and its circumstances varied, which could be expected if they were cut between age 3 and 4. Children have normally little remembrance of a particular event before the age of 3, but will usually be able to recall some events before the age of 4, reference number 76 and 77.

10. Please add the Ethical committee approval number both for the Norwegian regional ethics committee and the Gambian government/MRC joint ethics committee.

Response: Done. L254-255.

11. L118: what are the contents of physical complications?

Response: The content of physical complications is added in the method section, page 5 and 6, L160-166, and the result section L299-304.

12. Table 2: Perhaps some of the current physical complications are related to past complications (immediate physical complications)? If so, the author has to remove one of the variables from “immediate physical complications” and “current physical complications”, as there should be an overlap.

Response: We did examine for interactions between the immediate and current physical complications in the adjusted analyses as reported on page 8. We have added information that is more explicit; see statistical analyses, page 8, L243-244, and “note”, table 2.

13. Table 2: Why did the author not adjust with the variables “help received” in the analysis?

Response: We did adjust for “help received” in the analyses, but these variables did not show any associations with the outcome in the adjusted analyses. The variables adjusted for are given in the “note” below table 2.

14. L164-168: How did the author select those variables? Please describe the criteria of the selection to justify that the author did not pick them intentionally.

Response: The background variables selected (age, number of siblings, ethnic group affiliation and age in which the FGC procedure was conducted) were those included in the questionnaire, i.e. we used all background information available and did not omit any such variables. Unfortunately, other background information such as SES, marital status and educational level were not included in the questionnaire, limitation section, L529-530.

15. L249-251: “more than 50% had a fever”, “more than 90% of the girls had complications”. Are they Gambian prevalences or worldwide prevalences?

Response: The prevalence’s of immediate physical complications reported are those found in this sample of 12-year-old Gambian girls. We have discussed the high prevalence and compared with results from other studies further in the discussion section: L393-409. 

1. Is the manuscript technically sound, and do the data support the conclusions?

Reviewer #1: Yes

Reviewer #2: Yes

2. Has the statistical analysis been performed appropriately and rigorously? 

Reviewer #1: Yes

Reviewer #2: No

Response: revised according to comments by reviewer 2, statistics, L 247-250, and results, table 3.

3. Have the authors made all data underlying the findings in their manuscript fully available?

The PLOS Data policy <http://www.plosone.org/static/policies.action#sharing> requires authors to make all data underlying the findings described in their manuscript fully available without restriction, with rare exception (please refer to the Data Availability Statement in the manuscript PDF file). The data should be provided as part of the manuscript or its supporting information, or deposited to a public repository. For example, in addition to summary statistics, the data points behind means, medians and variance measures should be available. If there are restrictions on publicly sharing data—e.g. participant privacy or use of data from a third party—those must be specified.

Reviewer #1: No

Reviewer #2: Yes

Response: See the response to the academic editor about the lack of approval for data sharing.

4. Is the manuscript presented in an intelligible fashion and written in standard English?

Reviewer #1: Yes

Reviewer #2: Yes

Response to reviewer #1

Reviewer #1: The authors present findings for the research study titled “Physical complications following female genital cutting affect the mental health of 12-year-old Gambian girls: A community-based cross-sectional study”. Although the findings add towards the evidence of the impact of FGC on mental health on girls in Gambia, I find gaps in methods for it to be published as is. The following are the major and minor comments towards the manuscript.

Title: Physical complications following female genital cutting affect the mental health 1 of 12-year-old Gambian girls: A community-based cross-sectional study. I suggest modification to the title because cross-sectional studies do not depict causality.

Response: Done.

The Abstract is well written and reads well.

Introduction

i.Include a description of who performs the FGC on girls in Gambia Methods Section i. It will be helpful to describe in details the study sites/settings especially the schools

Response: We have included a section of who performs the FGC on girls in the Gambia, L68. 

The questionnaire included a question about who performed the cutting , which we have looked into upon your request. We found that among those who remembered the event, the procedure was reported being performed by traditional circumcisers in 97.6%, while only 2 girls reported that a health professional (doctor, midwife or nurse) performed the procedure. I am not quite sure how this information fits into the scope of this paper. One possibility could be to examine its associations with immediate or current complications, but, again, we have too few reports on medicalization of the procedure, which make it difficult to draw any conclusions.

The study sites were the schools in the Kanifing region (Education Region One of The Gambia). These schools were chosen in collaboration with the Regional Education Directorate, Region one, who gave a list of 26 schools, and the Department of Public and Environmental Health. School of Medicine and Allied sciences at the University of the Gambia, who visited all schools, gave information about the study and invited them to participate. This is a densely populated urban area in the Gambia, have an ethnic mixed community and the highest school enrollment rates.

The study settings in the schools are described in the method section, L148-151.

Further description of the education system in the Gambia described in the next comment.

 ii. Describe in details the characteristics of education system in Gambia, especially because interviews were conducted in English for girls 12 years old?

Response: further information on the education system has been added L121-124.

The Gambia is a former British colony that gained full independence in 1965. The country has a state educational system inherited from the British. The English language is still the official language for government and education. It is quite likely that many of the girls in 5th and 6th grade are quite fluent in English language (see reference below). Besides, as part of the selection procedure, this study excluded girls who did not speak English language well enough. As described in the method section, we also explained enquiries and uncertainties in the local language. Parts of the training for the interviewers focused on this, where a professor at the university, especially skilled in the various local languages, identified and translated English words and concepts that potentially could cause problems understanding into local language. 

iii. How did the authors obtain information regarding FGC status and the complications they suffered then, as well as current complications?

Response: The information regarding FGC status and the immediate and current complications they suffered from was obtained by the structured interview. We have included more details on the specific questions pertaining to FGC in the method section, page 6, L160-166..

iv. How would a girl 12 years who was cut may be at 1 month or when younger than 4 years old remember the incidence?

Response: Children are able to remember an incidence, such as cutting, from the age of three, see reference number 70 and 71 (Manuscript). As a proportion of the girls who were cut before the age of four (n = 74) were able to give information about the procedure and it’s circumstances (between 82-86 girls), they might have undergone the procedure between the age of 3 and 4, but we have not data to verify this assumption. The number of girls who remembered the circumstances pertaining to the FGC procedure varied for the various questions, which make sense as there are some uncertainty associated with memory for this age group, as well as individual differences. However, we chose to use the proportion who gave information of their physical complications following the procedure (n = 82), which is a conservative estimate as more girls recalled what kind of help they received afterwards.

v. Did the authors had methods of validating the information presented by the girls

Response: Yes, but only partly as no clinical examination was done. 

We have added information about validation of information, the method section, L144-148 and L176-178 AND in in the Discussion section/ Limitations, L545-550. 

vi. How did you address recall bias that is widely documented in regard to FGC

Response: We have added information about validation of information, the method section, L144-148 and L176-178 AND in in the Discussion section/ Limitations, L545-550

We addressed recall bias by thoroughly reviewing of our data looking for inconsistencies in the answers given by the girls on the different questions pertaining to FGC. Some inconsistencies were found among the 16% of girls who were cut before the age of four in that they recalled some of the circumstances, but not all. 

The questionnaire included an option to answer if they did not know whether they were cut. None of the included girls were uncertain of their FGC status, which may, given the topic of this project, reflect that the issue was discussed and clarified by parents beforehand. 

vii. How was sampling done?

Response: sampling was done by female teachers or head masters at school. Girls in grade 6 were gathered in groups, given information about the study, volunteered and picked out by teachers if they were considered being fluent in English language. L84-88 and L131-138.

viii. It would be helpful to describe what the traditional healers and religious persons did in supporting of the girl following the FGC-related complications. Again how would the young girl recall?

Response: Recollection varied as described earlier, but it’s reasonable to believe that they were most children have some degree of remembrance from the age of three. We have looked into the different kind of traditional help upon your request and added information, L170-176.

Results

i.Page 10 line 215-218: Receiving traditional help from the circumciser or a religious leader following immediate complications was not associated with any mental health outcomes, while the girls taken to a healthcare facility had a lower risk for depression. This is an important point but under described in the discussion. Could it be the girls would think by being taken back to the healers they would again be done something more harmful? In the absence of information on who performs the cut I can only speculate.

Response: We have added information, L170-176.

Discussion

i. It will be helpful to the shorten it

Response: Done.

Limitations

i. Shorten the limitation to a few sentences

Response: Done

 ii. The conclusion draws from the result but can benefit from a re-look

Response: Done

Response to Reviewer #2

 Introduction

• It is better to focus on the topic

Response: we regret that it may seem as if we have not done so and I am not sure to accommodate. The association between physical complications and mental health for young females who have undergone FGC is the topic of this paper. The introduction briefly describe relevant issues pertaining to FGC and physical complications (L47-57), studies on FGC and mental health outcomes (L58-62), as well as the link between physical injuries, pain and mental health (L63-66) with the intention of introduce the reader to these concepts and contexts. 

Methods

• Please mention the inclusion and exclusion criteria 

Response: Done. L84-90.

• It is not clear how the final sample size is determined 

Response: For this present paper, we included only those who had undergone FGC (n =134), which should be a representative sample and sufficient subset of the sample in order to investigate the impact of physical complication on mental health outcomes (i.e. more than 10 cases per explanatory variable in the regression analyses). 

However, the sample in the present paper is based on a project where the sample size calculations, which gave an estimate of about 338 cases, by using 95% CI, Margin of Error (SE), 5%. https://www.surveymonkey.com/mp/sample-size-calculator/ The sampling frame included a study population of 12-year-old female students in 28 lower basic schools of an ethnic diverge region in the Gambia (eligible, n = 2747). Each school was considered to be a category and the quota of each school was considered to be 10-14 participants, randomly selected, which should be a representative subset for the target population (i.e. urban area, lower basic school, different ethnic groups, FGC status, various measures on mental health outcomes). 

However, some schools refused to participate, and in some sites girls or their parents declined, which gave a total of 251 participants from 23 different schools included in this project. Based on the inclusion and exclusion criteria, L85-91, the final sample included 134 girls. 

• It is not clear how the schools were selected, Further description of the steps involved in the selection of the schools is needed 

Response: Done L121-124.

The Gambia has six school regions including Greater Banjul and Kanifing Municipal Council. These two school regions were chosen because more than 80% of all girls going to school resided in these two regions. Further, the regions also has a more mixed ethnic student population compared to other regions. The collaborating institution – University of the Gambia, Department of Public & Environmental Health - is also closer to these regions, which made it possible to conduct the study within planned budget.

The project worked with the Ministry of Basic and Secondary Education (MoBSE) to obtain a list of all primary schools in these two school regions. Twenty-six of the 28 schools in the region were visited for a consultation with the headmaster to inform them of the study plan, procedures and timeline. Three schools were not willing to participate, so 23 schools were part of the study.

• Why do the authors select 12 years old girls as a study subject 

Response: The project wanted to target preadolescent girls because they are more likely to have the cutting procedure more freshly in memory than at an older age. Further, we assume that they are not yet sexually active, thus, less prone to be influenced by the potential sexual problems described by several studies. At this young age, we considered a potential to minimize disturbing factors such as accumulated memories of stressful situations other that FGC, which of course in addition, could affect psychological health and well-being.

• Pretest might be important for such questionnaire , have you done a pretest 

Response: Yes, we have added information on the pilot-testing, L125-126.

Pretest of the questionnaire was done during the two first weeks of January 2018. Three out of the 12 female students (data-collectors) from the University of Brikama conducted a pilot-testing of the interview, where 6 12-year-old girls from one of the lower basic schools were interviewed. 

• In the measures subsection section ,the authors pointed out that “we asked the girls whether or not they had been undergone FGC”

There is no need to ask this if the samples have undergone FGM. What is the intention of asking this question?

Response: We fully agree in this comment as the final sample included only cut girls. It’s a lag due to the original recruitments described in the method section, page 4, i.e. the participants in this study were drawn from a larger sample, of which in advance of the recruitments didn’t have the information about FGC status.

• There is no need to do mean , median, SD, IQR for the same variable, It depends on the presence of normal distribution, please check the normal distribution and do the required analysis accordingly

Response: Both the SMFQ-score and the SCL-5 scores are skewed and clustered to the left. The Kolmogorov Smirnov test of normality for both SMFQ and SCL-5 are significant, i.e. not a normal distribution. The medians and IQRs were presented for this reason and to clarify the background of variables used in table 2, the Poisson regression analyses. 

see response to next comment 

• Please discuss about whether the data fulfilled the assumptions of t-test before conducting it 

Response: Thank you for making us aware of this. The distribution of both these variables are skewed and we have now corrected the analytic strategy and performed Mann-Whitney U Tests for table 3. However, the results from Mann-Whitney U-tests are similar to the T-tests, which probably reflect the fact that T-tests are quite robust even when the distribution is skewed.

Amended in the manuscript, L231, and L247-250, as well as table 3.

Results

• It is not presented correctly, The concept mentioned in table 1 is written all over the result section

Response: We have deleted redundant text in the result section that is also presented in the tables, page 10, 11 and 12.

 • The authors try to mention two separate concepts in table one 

Response: We regret this, but do have a problem understanding this comment and what actions that are suggested

• The minimum and maximum values can be mentioned as single variable (range) 

Response: Done. table 1.

• 52 of the subjects didn't remember the complication, then how did 6 of them remember whether they received help or not 

Response: see answer to #reviewer 1, comment v and iv.

• it might be wrong to measure the presence of complication based on subjective data, Did the authors believe that the subjective data accurately represent the study objective

Response: We are aware of the risk for information bias based on subjective data on FGC as mentioned in the limitation section, L544-551.

• Line 197,the authors mentioned that nearly all the girls who remembered the event and its circumstances (n = 86, 64%) while the value mentioned in table 1 is 82(61.2)

Response: see response to comment iv from # reviewer 1.

• the association with number of siblings is not clear, it is better to elaborate it in the discussion section

Response: We thank the reviewer for this comment. We have added information on this in the discussion section, L386-387 AND L439-442.. 

 • what is the need of assessing the association of mandinka ethnic group with the outcome variables, what will be the recommendation if it showed the association 

Response: I am not sure if such an association would have generated any particular recommendations, but if the extent of reported physical complications was associated with ethnic group affiliation, new hypothesis might have been generated though these data, for instance type and age of the procedure. The literature on what age FGC is performed among the different ethnic groups in the Gambia is sparse. Hypothetically, these aspects could affect both physical health and complications, and in turn, mental health outcomes. 

• Why do the authors consider age of 4 in comparing the mean value difference in the levels of depressive symptoms or psychological distress

Response: See the response to comment 9 from the academic editor. 

Discussion 

• It seems a literature review 

Response: we regret that could be perceived as such 

• Please delete or elaborate concepts that are not presented in the result section. The discussion section should be written in line with the results 

Response: We have deleted the following concepts not presented in the results: i) page 17, L467, and ii) page 18, L605.

• Please try to compare the findings from various studies with your study References 

Response: We believe that we have compared findings from various studies with those from our study: i) immediate physical complications, L483-499, Ii) current physical complications, 500-508. Iii) Psychological impact of FGC in children and adolescents, L514-528 AND 533-544. Iv)Low utilization of healthcare, L545-556. v) If the age at which FGC procedure is performed, i.e. if recollection of this event affect mental health and well-being, L558-605. 

• Many references are old

Response: Some old references were relevant for this the topic of this paper. We have kept the references for this reason. 

---

## [Decision Letter · Decision Letter 1]

7 Jan 2021

The association between Physical complications following female genital cutting and the mental health of 12-year-old Gambian girls: A community-based cross-sectional study

PONE-D-20-03360R1

Dear Dr. Bendiksen,

We’re pleased to inform you that your manuscript has been judged scientifically suitable for publication and will be formally accepted for publication once it meets all outstanding technical requirements.

Kind regards,

Kimiyo Kikuchi

Academic Editor

PLOS ONE

Additional Editor Comments (optional):

Reviewers' comments:

Reviewer's Responses to Questions

**Comments to the Author**

1. If the authors have adequately addressed your comments raised in a previous round of review and you feel that this manuscript is now acceptable for publication, you may indicate that here to bypass the “Comments to the Author” section, enter your conflict of interest statement in the “Confidential to Editor” section, and submit your "Accept" recommendation.

Reviewer #1: All comments have been addressed

Reviewer #2: All comments have been addressed

2. Is the manuscript technically sound, and do the data support the conclusions?

Reviewer #1: Yes

Reviewer #2: Yes

3. Has the statistical analysis been performed appropriately and rigorously? 

Reviewer #1: Yes

Reviewer #2: Yes

4. Have the authors made all data underlying the findings in their manuscript fully available?

Reviewer #1: Yes

Reviewer #2: Yes

5. Is the manuscript presented in an intelligible fashion and written in standard English?

Reviewer #1: Yes

Reviewer #2: Yes

6. Review Comments to the Author

Reviewer #1: The authors present findings for the research study titled “The association between Physical complications following female genital cutting and the mental health of 12-year-old Gambian girls: A community-based cross sectional study”. The findings add towards the evidence of the impact of FGM on mental health on girls in Gambia.

The manuscript has greatly and significantly improved, while the authors have comprehensively addressed the comments that were raised.

I am satisfied with the improvement to the manuscript and responses by the authors.

Reviewer #2: The authors have made a major change. The author has addressed all my comments. I don't have additional comment.

7. PLOS authors have the option to publish the peer review history of their article (what does this mean?). If published, this will include your full peer review and any attached files.

Reviewer #1: No

Reviewer #2: **Yes: **Robera Olana Fite

---

## [Editor Report · Acceptance letter]

12 Jan 2021

PONE-D-20-03360R1 

The association between Physical complications following female genital cutting and the mental health of 12-year-old Gambian girls: A community-based cross-sectional study  

Dear Dr. Bendiksen:

I'm pleased to inform you that your manuscript has been deemed suitable for publication in PLOS ONE. Congratulations! Your manuscript is now with our production department. 

Kind regards, 

on behalf of

Dr. Kimiyo Kikuchi 

Academic Editor

PLOS ONE